# A Dynamic Service Placement Based on Deep Reinforcement Learning in Mobile Edge Computing

**Shuaibing Lu** [1,†] ⓘ**, Jie Wu** [2,*,†]**, Jiamei Shi** [1,†]**, Pengfan Lu** [1,†]**, Juan Fang** [1,†] ⓘ **and Haiming Liu** [3,†]

1 Faculty of Information Technology, Beijing University of Technology, Beijing 100124, China; lushuaibing@bjut.edu.cn (S.L.); shijiamei@emails.bjut.edu.cn (J.S.); lu_peng_fan@emails.bjut.edu.cn (P.L.); fangjuan@bjut.edu.cn (J.F.)
2 Center for Networked Computing, Temple University, Philadelphia, PA 19122, USA
3 School of Software Engineering, Beijing Jiaotong University, Beijing 100044, China; liuhaiming@bjtu.edu.cn
* Correspondence: jiewu@temple.edu
† These authors contributed equally to this work.

**Abstract:** Mobile edge computing is an emerging paradigm that supplies computation, storage, and networking resources between end devices and traditional cloud data centers. With increased investment of resources, users demand a higher quality-of-service (QoS). However, it is nontrivial to maintain service performance under the erratic activities of end-users. In this paper, we focus on the service placement problem under the continuous provisioning scenario in mobile edge computing for multiple mobile users. We propose a novel dynamic placement framework based on deep reinforcement learning (DSP-DRL) to optimize the total delay without overwhelming the constraints on physical resources and operational costs. In the learning framework, we propose a new migration conflicting resolution mechanism to avoid the invalid state in the decision module. We first formulate the service placement under the migration confliction into a mixed-integer linear programming (MILP) problem. Then, we propose a new migration conflict resolution mechanism to avoid the invalid state and approximate the policy in the decision modular according to the introduced migration feasibility factor. Extensive evaluations demonstrate that the proposed dynamic service placement framework outperforms baselines in terms of efficiency and overall latency.

**Keywords:** dynamic service placement; delay optimization; cost efficiency; mobile edge computing

## 1. Introduction

The evolution of the Internet of Things (IoT) promotes the development of our society, which requires highly scalable infrastructure to provide proper services for diverse applications adaptively [1]. As a promising framework, mobile edge computing (MEC) supports the exponential growth of emerging technologies, such as online interactive games, augmented reality, real-time monitoring, and so on by pushing the computation, storage, and networking resources to the base stations. However, users demand a higher quality-of-service (QoS) with increased investment of resources, and it is nontrivial to maintain service performance under the erratic activities of end-users and limited capacities. In this paper, we study the service placement problem by minimizing the total delay of multiple users under the long-term cost constraint.

### 1.1. Motivation and Challenges

An illustration of the dynamic service placement problem is shown in Figure 1 to represent the unique challenges under this problem. (i). Since there are no restrictions on the locations of services, where these services are placed so that they can reach better utilization on the physical resources of edge servers include the aspects on the computing, communication, and storage in MEC is nontrivial. For example, suppose that the computing capacity of edge server $m_2$ in area 2 is much higher than others with lower storage. When the movement trajectories of users overlap with the areas nearby $m_2$, the services that

correspond to these users are expected to be placed at a server that is close to them and has better performance. However, it is obvious that the available storage capacity of $m_2$ cannot satisfy the requirements of all users. Therefore, how the system can deal with the services that attempt to migrate over requesting high computing capacity with limited storage resources is important. (ii). The services serve users one-to-one, and the activities of users are erratic. It is nontrivial to find an efficient strategy that adapts the erratic movements by considering minimizing the total delay under the cost constraint. As shown in Figure 1, we suppose that users in areas 1, 3, and 4 are on the move at time slot $t$. One of the simple solutions to maintain performance is to migrate services in order to follow users, which produces lower latency. However, frequent service migration will bring additional traffic load in the backhaul network and higher operational costs. Therefore, it is challenging to deal with the services that can realize dynamic adaptation with low latency under limited cost.

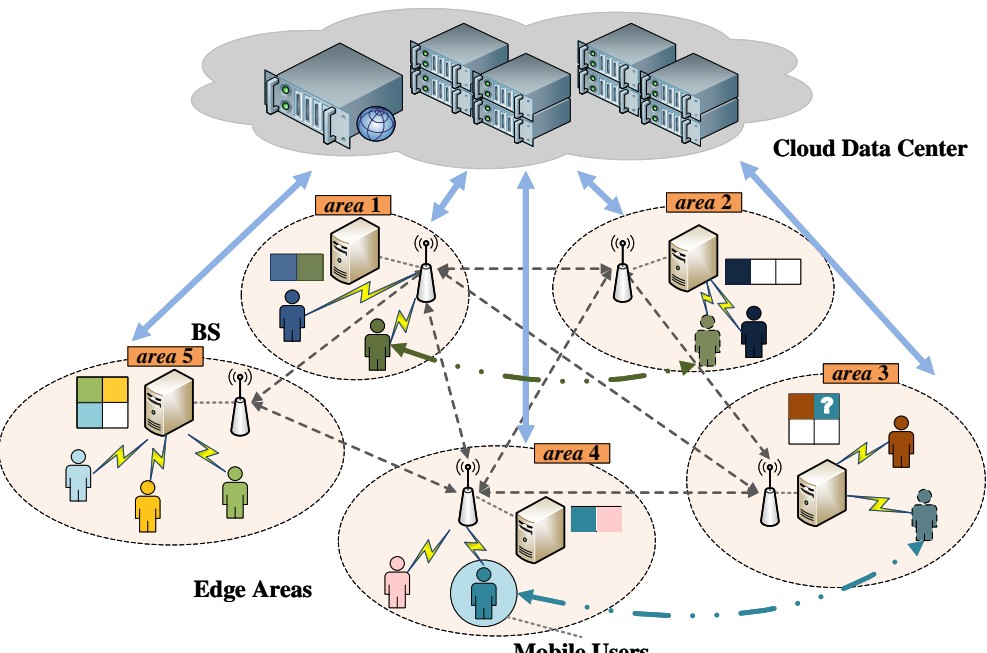

**Figure 1.** An illustration of the dynamic service placement in mobile edge computing.

*1.2. Contributions and Paper Organization*

In this paper, we introduce a novel dynamic placement framework based on deep reinforcement learning (DSP-DRL) to optimize the QoS for users under the constraints on physical resources and operational costs. Our contributions can be summarized as follows:

- We investigate the service placement problem in mobile edge computing with multiple users, and we propose to minimize the total delay of users by considering the limitation on physical resources and cost.
- We propose a decentralized dynamic placement framework based on the deep reinforcement learning (DSP-DRL) by introducing the migration conflict resolution mechanism during the learning process to maintain the service performance for users. We formulate the service placement under the migration conflict into a mixed-integer linear programming (MILP) problem. Then, we propose a migration conflict resolution mechanism to avoid the invalid state and approximate the policy in the decision modular according to the migration feasibility factor.
- Extensive evaluations demonstrate that the proposed dynamic service placement framework outperforms baselines in terms of efficiency and overall latency.

The remainder of this paper is organized as follows. Section 2 surveys related works. Section 3 describes the model and then formulates the problem. Section 4 investigates the dynamic service placement framework based on deep reinforcement learning. Section 5 includes the experiments. Finally, Section 6 concludes the paper.

## 2. Related Work

The concept of mobile edge computing is introduced to extend the cloud paradigm, which enables a new breed of services and applications. It provides a service environment closer to both users and IoT devices by deploying several mobile edge servers. Service placement is a well-investigated problem in mobile edge computing that advocates for providing service offering at the users' side [2,3]. Various works have studied different aspects of this problem. A subset of existing work in this area relates to improving the utilities and reducing the operational cost. Ning et al. [4] propose a dynamic storage-stable service placement strategy by using the Lyapunov optimization method to maximize the system utility while striking a balance between the overhead and stability. Pasteris et al. [5] focus on the problem of service placement by considering a heterogeneous mobile edge computing system, and they propose a deterministic approximation algorithm to maximize the total revenue. Chen et al. [6] propose an efficient decentralized algorithm by exploiting the graph coloring on the small cell network for performing collaborative service placement in order to optimize the utility of operators. Yu et al. [7] investigate the collaborative service placement problem in mobile edge computing by proposing an efficient decentralized algorithm based on the matching theory. They try to minimize the traffic load to realize a high utilization on computing and radio resources. Gu et al. [8] focus on the layer-aware service placement and request scheduling problem, and they design an iterative greedy algorithm by formulating it into an optimization problem with approximate submodularity. In addition, quite a few works have been carried out on optimizing the quality-of-service (QoS). Xu et al. [9] tackled it by proposing a trust-oriented IoT service placement method for smart cities in edge computing, and they try to optimize the execution performance with privacy preservation. Maia et al. [10] formulate the load distribution and placement problem as integer nonlinear programming, and they try to minimize the potential violation to improve the QoS by using the genetic algorithm. Fu et al. [11] propose a runtime system that effectively deploys user-facing services in a cloud-edge continuum to ensure the QoS by jointly considering communication, contention, and load condition. However, these works ignore the coupling relationship between service performance and operational cost caused by users' erratic movements.

In response to the challenge on users' mobilities across multiple timescales, some works are based on service migration in mobile edge computing. There are a few works that assume that the user mobility follows a Markovian process and apply the Markov Decision Process (MDP) technique. Wang et al. [12] formulate the service migration problem with minimum cost as an MDP and propose a new algorithm for computing the optimal solution that is significantly faster than traditional methods based on standard value or policy iteration. Gao et al. [13] jointly optimize the network selection and service placement to improve the QoS by considering switching and communication delay, and they propose to dynamically place and migrate the services according to the mobility of users by introducing an iteration-based algorithm. Tao et al. [14] study the mobile edge service performance optimization problem by applying the Lyapunov optimization. They design an approximation algorithm based on Markov approximation under long-term cost budget constraint. Since the characteristics of mobile users are moving without a priori knowledge, some researchers introduce deep reinforcement learning.

Rui et al. [15] propose a novel service migration method based on state adaptation and deep reinforcement learning to overcome network failures, and they use the satisfiability modulo theory to solve the candidate space of migration policies. Liu et al. [16,17] design a reinforcement learning-based framework by using a deep Q-network for a single user service migration system, which was realized to choose the optimal migration strategy in

edge computing. Yuan et al. [18] study the service migration and mobility optimization problem by proposing a two-branch convolution-based deep Q-network to maximize the composite utility. However, these works make decisions by calculating the *Q*-value of the state and action, which are not precise, since the trajectories of mobile users are uncertain and dynamic in a timescale. Pan et al. [19] develop a novel hierarchical reinforcement pricing by capturing both spatial and temporal dependencies based on the deep deterministic policy gradient (DDPG) [20]. Wei et al. [21] consider a more practice-relevant scenario in which multiple mobile users generally have a small size and can be easily moved around and distributed at different edge servers for processing, and they propose a reinforcement learning-based algorithm that leverages the learning capability of DDPG. However, these works do not take into account the problems of resource limitation and migration conflict under the case that multiple users own similar activities.

In this paper, we study the service placement problem under the continuous provisioning scenario in mobile edge computing. Our objective is to minimize the total delay under the physical resources by considering maintaining service performance under the erratic activities of multiple users.

## 3. Model and Problem Formulation

In this paper, we study the service placement problem in mobile edge computing while jointly considering the QoS of users and cost of service operators. Our objective is to minimize the total delay of users and maintain the performance without overwhelming the constraints on physical resources and operational cost. In this section, we start with the descriptions of the system model and the QoS model. The problem is also formulated.

### 3.1. System Model

First, we are given a substrate distribution of MEC nodes $M = \{m_j\}$ that are supported by the network operator. Each MEC node is attached to a base station with limited computing and storage capacities, where $R^c_{m_j}$ denotes the computing capacity of $m_j$, and $R^s_{m_j}$ denotes the storage capacity of $m_j$. We use a set $U = \{u_i\}$ to denote the users with mobilities that are served by the MEC nodes. The users that subscribe to the services from the MEC operators are distributed over the coverage region of the base station. To better capture the users' mobilities, the system is assumed to operate in a slotted structure, and its timeline is discretized into time frame $t \in \mathbf{T} = \{0, 1, 2, ..., T\}$ [14]. At all discrete time slots, each mobile user sends a service request to the MEC node that can be accessed. We use $V$ to denote the set of services that are supported by the operators, where $V = \{v_h\}$. We assume that the services are deployed on the virtual machines, and each user can only be served by one service on the MEC. To simplify the description, we use color squares to represent the placed services. Each MEC has a service range shown in Figure 1. Here, we suppose that the capacities of MECs are heterogeneous, and their service ranges are different. We use light orange color circles with different sizes to represent the coverage ranges of each MEC. Let $x^j_{ih}(t) = 1$ denote user $u_i$ using the service $v_h$ which is placed on edge server $m_j$ at time slot $t$; otherwise, $x^j_{ih}(t) = 0$. For each MEC node, we use $\mathbb{V}_{m_j}$ to denote the set of services that are placed on edge server $m_j$, where $\mathbb{V}_{m_j} = \{v_h | m_j \leftarrow v_h\}$. We suppose that each service only serves one user at a time, and we use $\mathbb{U}(\mathbb{V}_{m_j})$ to denote the set of users that is served by the services in set $\mathbb{V}_{m_j}$. For the convenience of reference, we summarize the main notations throughout this paper in Table 1.

**Table 1.** List of main notations.

| Notation | Definition |
|:---:|:---:|
| $M$ | Set of MEC nodes, where $M = \{M_j\}$. |
| $U$ | Set of users, where $U = \{u_i\}$. |
| $V$ | Set of services, where $V = \{v_h\}$. |
| $\mathbb{V}_{m_j}$ | Set of services placed on edge server $m_j$. |
| $\mathbb{U}(\mathbb{V}_{m_j})$ | Set of users served by the services in set $\mathbb{V}_{m_j}$. |
| $x_{ih}^j(t)$ | A boolean variable that indicates $v_h$ serving $u_i$ on edge server $m_j$ at time slot $t$. |
| $A_i(t)$ | The amount of required computing resource of $u_i$ at time slot $t$. |
| $D_{u_i}^c(t)$ | The computing delay of $u_i$. |
| $D_{u_i}^l(t)$ | The communication delay of $u_i$. |
| $D_{u_i}^u(t)$ | Updating delay of $u_i$ during the dynamic migration. |
| $t_{u_i,m_j}(t)$ | Maximum transmission rate between $u_i$ and $m_j$. |
| $b_{u_i,m_j}(t)$ | Channel bandwidth of link between $u_i$ and $m_j$. |
| $p_{u_i,m_j}(t)$ | Physical distance between $u_i$ and $m_j$. |
| $R_{m_j}^s$ | The storage capacity of $m_j$. |
| $R_{m_j}^c$ | The computing capacity of $m_j$. |

*3.2. QoS Model*

3.2.1. Computing Delay

We use $D_{u_i}^c(t)$ to denote the computing delay of user $u_i$ at time slot $t$. Let $A_i(t)$ denote the amount of computing resource required by the service request of user $u_i$ at time slot $t$. In this paper, we consider that each user shares the computing resource of the MEC sever evenly [14,22]. Here, the computing resources are measured by the number of CPU cycles.

$$D_{u_i}^c(t) = \sum_{m_j \in M} \sum_{u_i \in U} \frac{x_{ih}^j(t) \cdot A_i(t)}{R_{m_j}^c} \tag{1}$$

3.2.2. Communication Delay

The communication delay occurs when the service is not placed in the user's area, which is determined by the data transmission and the network propagation. The network propagation is determined by the distance $p_{u_i,m_j}(t)$ between user $u_i$ and service $v_i$ placed on edge node $m_j$, such as hops [23]. Let $t_{u_i,m_j}$ denote the maximum transmission rate, where

$$t_{u_i,m_j}(t) = b_{u_i,m_j}(t) \cdot \log_2(1 + \frac{\tau \cdot g(u_i, m_j)}{N}) \tag{2}$$

We use $b_{u_i,m_j}$ to denote the channel bandwidth of the physical link, and $\tau$ denotes the transmission power of the local mobile device of $u_i$. Let $g(u_i, m_j)$ represent the channel gain between $u_i$ and MEC $m_j$, where $g(u_i, m_j) = 127 + 30 \cdot \log p_{u_i,m_j}(t)$ [24]. Let $N$ represent the noise power. The data transmission is determined by the bandwidth of the physical link $b_{u_i,m_j}$ and the data size of the request $d_{u_i}(t)$ when it passes through the network devices between the connected MEC node and the service provided one. Therefore, the communication delay is

$$D_{u_i}^l(t) = \sum_{m_j \in M} x_{ih}^j(t) \cdot \frac{d_{u_i}(t)}{t_{u_i,m_j}(t)} \tag{3}$$

### 3.2.3. Updating Delay

Due to the mobilities of users, it is inefficient to keep the locations of services unchanged all the time, which will increase the communication delay of users. Thus, we consider optimizing the user experience via dynamically migrating the services. We define a boolean variable $\alpha(v_i)$ to denote whether the service $v_i$ that is serving user $u_i$ is under the migration or toggling state. $Y(v_i)$ is the updating delay of service $v_i$, which includes service profiles transmission, rebooting software resources, and so on [13]. The updating delay $D_{u_i}^u(t)$ of user $u_i$ is defined as

$$D_{u_i}^u(t) = \alpha(v_i)(t) \cdot Y(v_i) \tag{4}$$

### 3.3. Problem Formulation

In this paper, we consider achieving the dynamic service placement by minimizing the total delay of multiple mobile users under the physical resource and cost constraints. We suppose that the cost during the dynamic service placement process is produced by the migration of services across edge servers. In order to satisfy the quality-of-service (QoS) requirements of users under the erratic movement, the service should be dynamically migrated to adapt to the users' mobility; however, the resulting cost for the operators will be excessive. Let $\rho$ denote the unit cost of $v_i$ during the service migration, and the cost is defined as

$$C_{m_i,m_j}^h(t) = \rho \cdot D_{u_i}^u(t) \tag{5}$$

Moreover, we use $\Gamma$ to represent the higher bound of the maximum total cost that is afforded by the operators. The problem formulation is shown as follows:

$$\text{minimize} \sum_{t=0}^{T} \sum_{j=1}^{|M|} \sum_{i=1}^{|U|} D_{u_i}^c(t) + D_{u_i}^l(t) + D_{u_i}^u(t) \tag{6}$$

$$\text{s.t.} \quad \sum_{t=0}^{T} \sum_{h=1}^{|V|} C_{m_i,m_j}^h(t) \le \Gamma, \tag{7}$$

$$\sum_{v_h \in \mathbb{V}_{m_j}} |v_h| \le R_{m_i}^s, \quad \sum_{u_i \in \mathbb{U}(\mathbb{V}_{m_j})} A_i(t) \le R_{m_i}^c, \forall j \in M, \tag{8}$$

$$x_{ih}^j \in \{0,1\}, \forall i \in U, \forall h \in V. \tag{9}$$

Our objective is to minimize the total delay of users in set $U$ during a continuous time period in Equation (6). Equations (7)–(9) are the constraints. Equation (7) states the cost constraint, which means that the total cost of the provided services cannot exceed the threshold $\Gamma$. Equation (8) states the constraint on the physical resources, where the services that are placed on edge sever $m_i$ cannot exceed the storage $R_{m_i}^s$, and the amount of computing resources required by the service request of $u_i$ cannot exceed the computing capacity $R_{m_i}^c$. Equation (9) states whether user $u_i$ uses service $v_h$ at time slot $t$.

## 4. Dynamic Service Placement Framework Based on Deep Reinforcement Learning

In this section, we show the detail of our novel decentralized dynamic service placement framework based on the deep reinforcement learning approach to realize the lower delay under the constraints on physical resources and costs. There are two networks (main network and target network) in our framework. In the main network, the critic network is used to output real-time actions for actors to implement in reality, while the actor network is used to update the value in the network system. In the target network, they are all outputting the value of this state, but the inputs are different. The critic network will analyze the action from the actor network plus the observation value of the state, and the

actor network will take the actor at that time. Figure 2 shows the overall architecture of the DSP-DRL framework.

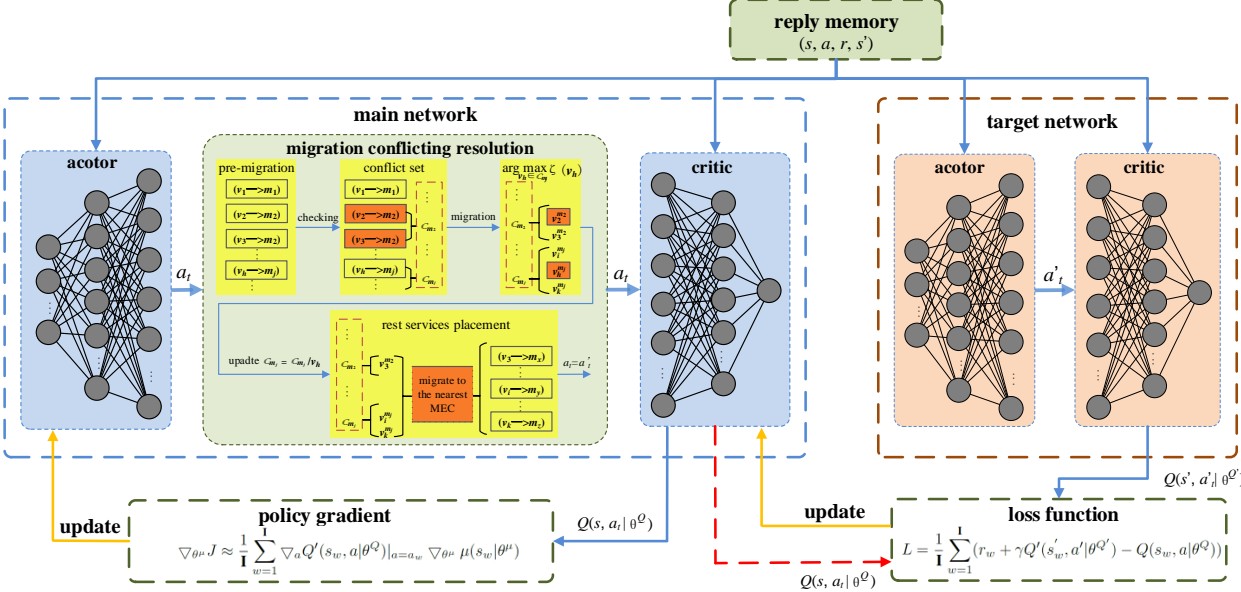

**Figure 2.** The overview of the DSP-DRL framework.

### 4.1. Deep Reinforcement Learning Formulation

Since the decision making during dynamic service placement is a stochastic optimization, our framework is studied based on the deep deterministic policy gradient (DDPG) algorithm [25]. In this paper, the objective of the agent is to realize dynamic service placement for multiple mobile users while minimizing the total delay. We first summarize the state and action spaces, reward function, and the state transition policy that are used in our reinforcement learning framework. In order to describe the environment of edge servers and mobile users for the agent concisely and correctly, the state space includes the knowledge of services placed on the edge servers and the status of users that are supplied by these services. To that end, the state is designed as follows.

**Definition 1 (State).** *The state $s_t$ describes the environment of the edge network, which is a vector consisting of $s_t = [\mathbf{r}_t, \hat{\mathbf{u}}_t]$. $\mathbf{r}_t = (r_1(t), r_2(t), ..., r_j(t), ..., r_m(t))$ is the vector of rest storages, where $r_j(t)$ denotes the rest storage on edge server $m_j$. $\hat{\mathbf{u}}_t = (\hat{u}_1(t), \hat{u}_2(t), ..., \hat{u}_i(t), ..., \hat{u}_n(t))$ is the vector of positions on each users' trajectories, where $\hat{u}_i(t)$ denotes the position of $u_i$ at time slot $t$.*

We consider that the services on each edge server make decentralized decisions according to the trajectories of mobile users by training the agent. The action $a_t$ is designed as follows:

**Definition 2 (Action).** *The action space $a_t = [\hat{\mathbf{m}}_1, \hat{\mathbf{m}}_2, ..., \hat{\mathbf{m}}_h, ..., \hat{\mathbf{m}}_n]_t$ is the migration action, where $\hat{\mathbf{m}}_h(t) = [\hat{m}_h(t)^-, \hat{m}_h(t)^+]$ denotes the alternative range of edge servers during the migration process of service $v_h$ at time slot $t$.*

For each service, the alternative range of edge servers is represented by a continuous set of numbers $[\hat{m}_h(t)^-, \hat{m}_h(t)^+]$, where $\hat{m}_h(t)^-$ denotes the lower bound (minimum number of server) that can be selected during the service $v_h$'s migration, and $\hat{m}_h(t)^+$ denotes the upper bound (maximum number of servers).

Since our problem is an online learning process, the value of the reward cannot determine the final total delay for multiple mobile users in each time slot directly; however, it will drive their behaviors to obtain a better performance. In order to realize a decentralized dynamic service placement strategy, we minimize the total delay while completing the

processed tasks for the mobile users within a limited migration cost. Thus, here is the specific definition of the reward function.

**Definition 3** (**Reward**). *The reward r is measured by the average delay feedback of multiple mobile users comparing with* $r = \sum_{u \in U} \frac{\overline{z_{u_i}}(t) - z_{u_i}(t)}{|U|}$.

Here, $|U|$ denotes the total number of users and $z_{u_i}(t)$ denotes the total delay of multiple users according to the decisions by the deep neural network at time slot $t$. We use $\overline{z_{u_i}}(t)$ to denote the total delay that the service stays on the original edge server without migration.

### 4.2. Migration Conflicting Resolution Mechanism

For each service, the decisions are made depending on the observation of the environment from their own perspectives during each episode. However, there is no prior knowledge of the mobile edge computing system, which means the data size and trajectories are unknown to each server. Thus, the process is online and model-free. In order to maintain the service performance, the migration of services and users' activities are tightly coupled. Since the multiple users move erratically and autonomously, there will be a conflict between multiple services due to similar or overlapping users' trajectories during the learning process. We use the following example to illustrate this problem, which is shown in Figure 3. Suppose that the activities of users $u_1$, $u_4$, and $u_8$ are all around area 4 (written in red text) at the same time slot. In this case, the chosen services have a high probability of migrating to the same edge server $v_4$ in the learning framework. However, the rest of the storage can only afford one service, which creates a migration conflict.

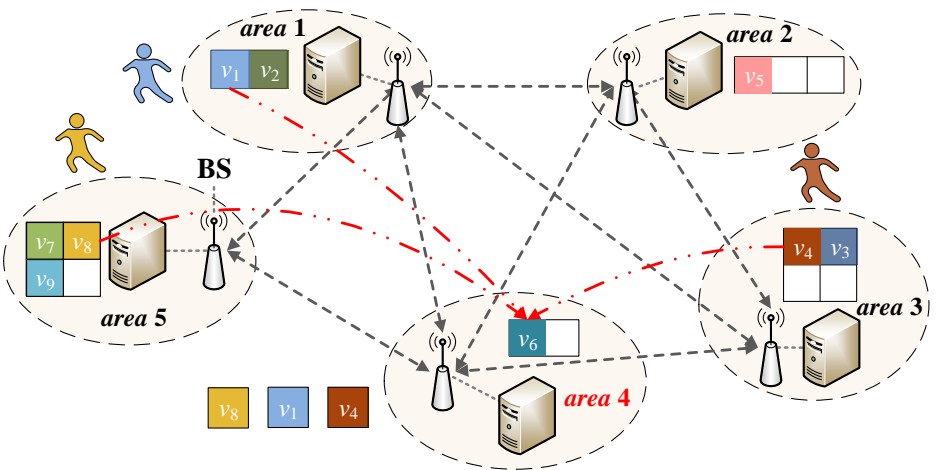

**Figure 3.** Services actions with migration conflict.

#### 4.2.1. Service Placement under Migration Conflict

We first formulate the service placement under migration conflict into a mixed-integer linear programming problem. As shown in the objective function in Equation (6), we aim to minimize the total delay of users in a time-varying period. We use $z_j(t)$ to denote the total delay of user $u_i$ at time slot $t$, where $z_{u_i}(t) = D_{u_i}^c(t) + D_{u_i}^l(t) + D_{u_i}^u(t)$. When service $v_i$, which is serving $u_i$, migrates successfully, the communication delay $D_{u_i}^l(t)$ will decrease; otherwise, $D_{u_i}^u(t) = 0$. Here, $\alpha(v_i)$ is a boolean variable that represents whether service $v_i$ migrates successfully. For the communication delay, the migration result produces two different values. We combine these two cases and transform the communication

delay into $D_{u_i}^l(t) = \alpha(v_i) \cdot D_{u_i}^{\prime l}(t) + (1 - \alpha(v_i))D_{u_i}^l(t)$. Thus, the total delay of user $u_i$ is transformed to

$$z_{u_i}(t) = D_{u_i}^c(t) + \alpha(v_i) \cdot (D_{u_i}^{\prime l}(t) + D_{u_i}^u(t)) + (1-\alpha(v_i))D_{u_i}^l(t). \tag{10}$$

For edge serve $m_j$, the total delay of users that are served by corresponding services is $Z_u^{m_j}(t) = \sum_{u_i \in \mathbb{U}(\mathbb{V}_{m_j})} z_{u_i}(t)$. When an edge server has a migration conflict, it means that multiple services choose it as a destination while its storage capacity cannot satisfy all services. The value of $Z_u^{m_j}(t)$ is divided into two parts; one is determined by services with conflicts, which are shown as follows:

$$Z_u^{\prime m_j}(t) = \sum_{u_i \in \mathbb{U}(\mathbb{C}_{m_j})} z_{u_i}(t). \tag{11}$$

Another one is produced by the placed services shown as follows:

$$Z_u^{\prime\prime m_j}(t) = \sum_{u_i \in \mathbb{U}(\mathbb{V}_{m_j} - \mathbb{C}_{m_j})} z_{u_i}(t). \tag{12}$$

Since the first part $Z_u^{\prime m_j}(t)$ is fixed, the optimization of $Z_u^{m_j}(t)$ will be transformed to the optimization of $Z_u^{\prime\prime m_j}(t)$. Therefore, the problem of minimizing total delay for the service placement under migration conflicts in time slot $t$ can be formulated as a mixed-integer linear programming problem as follows, which has been proven to be NP-hard [26].

$$\text{minimize} \sum_{k=1}^{|M|} Z_u^{\prime\prime m_j}(t) \tag{13}$$

$$\text{s.t.} \quad Z_u^{\prime\prime m_j}(t) \geq 0, \forall u_i \in U, \forall m_j \in M, \tag{14}$$

$$\alpha(v_i) \in \{0,1\}, \forall v_i \in V. \tag{15}$$

4.2.2. Migration Conflict Resolution Mechanism

We propose a new migration conflict resolution mechanism to avoid the invalid state and approximate the policy in the decision module. There are two main stages included in our resolution mechanism: stage one is to find the edge servers with conflicting services; stage two is to make migration decisions for the conflicting edge nodes. The details are shown in Algorithm 1. The input is action $a_t$ at time slot $t$, and the output is the updated action of service placement decisions while enabling conflict resolution for edge servers. According to the action $a_t$, which is produced during the learning process in time slot $t$, we do the pre-migration for each service $v_h \in \mathbf{V}$. After that, we check the status of each edge sever $m_j \in \mathbf{M}$ by calculating the total number of services $N_{m_j}$ pre-migrated to $m_j$. We compare the number of total requests $N_{m_j}$ with the storage $R(m_i)$ of destination server $m_j$. If $N_{m_j} > R(m_i)$, $m_j$ is a conflicting edge server. Otherwise, the migration to $m_j$ is successful. Based on that, we start to make migration decisions. For the conflicting edge server $m_j$, we first build conflict set $\mathbb{C}_{m_j} = \{v_h^{m_j}\}$, which is composed of all the services requesting to migrate on server $m_j$ at the same time. Then, we choose service $v_h$ in set $\mathbb{C}_{m_j}$ with maximum $\zeta(v_h)$. Here, we introduce a novel definition: migration feasibility factor.

**Definition 4 (migration feasibility factor).** *Let $\zeta(v_h)(t)$ indicate the migration feasibility factor of service $v_h \in \mathbb{C}_{m_j}$ at time slot $t$, where $\zeta(v_h)(t) = D_{u_h}^u(t) + \frac{\omega}{D_{u_h}^l(t)}$ and $\omega > 0$.*

We use $D_{u_i}^u(t)$ to denote the migration delay of the service $v_h$ that is serving user $u_h$ at time slot $t$, and we use $D_{u_h}^l(t)$ to denote the communication delay produced when service $v_h$ is not placed on the edge server within a user's area. These two parameters

are negatively correlated, which means that when the service migrates or is close to the users' area, the value of communication delay will be $D_{u_h}^l(t) = 0$ or less. Here, we use a constant $\varpi$ to adjust the relationship, where $\varpi > 0$. Therefore, the migration feasibility factor considers the impact of these two parameters on users' delays. In line 9, we update the conflict set with $\mathbb{C}_{m_j} = \mathbb{C}_{m_j}/v_h$. For the rest of the services in set $\mathbb{C}_{m_j}$, we migrate $v_{h'}$ to the nearest edge server that meets the storage resource requirements, as denoted in lines 10 to 12. In line 13, we record the current state of services placement in $a_t'$ and update the action $a_t = a_t'$. If the storage resources are adequate, there will be no conflict, which means that the services migrate to $m_j$ successfully. Finally, we will keep the original service placement decisions of action $a_t$ in line 15.

---

**Algorithm 1** Migration conflict resolution method

---

**Input:**    The action $a_t$ at time slot $t$;
**Output:**   The updated action $a_t$ of service placement decisions under the migration conflict edge servers;
 1: **for** each service $v_h \in \mathbf{V}$ **do**
 2:    Pre-migration according to $a_t$;
 3: **end for**
 4: **for** each edge server $m_j \in \mathbf{M}$ **do**
 5:    Calculate the total number of services $N_{m_j}$ pre-migrated to $m_j$;
 6:    **if** $N_{m_j} > R(m_j)$ **then**
 7:       Construct conflict set $\mathbb{C}_{m_j} = \{v_h^{m_j}\}$;
 8:       Choose service $v_h$ in set $\mathbb{C}_{m_j}$ with maximum $\zeta(v_h)$;
 9:       Update set $\mathbb{C}_{m_j} = \mathbb{C}_{m_j}/v_h$;
10:       **for** each service $v_{h'}$ in set $\mathbb{C}_{m_j}$ **do**
11:          Migrate $v_{h'}$ to the nearest edge server that meets the storage resource;
12:       **end for**
13:       Record current state of services placement in $a_t'$ and update $a_t = a_t'$;
14:    **else**
15:       Keep the original service placement decisions of action $a_t$;
16:    **end if**
17: **end for**

---

*4.3. Dynamic Service Placement Based on Deep Reinforcement Learning*

In this subsection, we propose a dynamic service placement strategy based on deep reinforcement learning. According to the characteristic of the decision-making process, our scheme studies are based on the deep deterministic policy gradient (DDPG) algorithm. The main idea is to use a deep reinforcement learning agent to perform the dynamic service placement of multiple mobile users to minimize the total delay.

The specific steps are shown in Algorithm 2. We use the sets of edge nodes $\mathbf{M}$, services $\mathbf{V}$, and users $\mathbf{U}$ as the input. The output is the dynamic service placement scheme $\mathbf{X}$. In lines 1 to 3, we initialize the preliminary parameters of the reinforcement learning agent, which includes the main network, the target network, and the replay buffer. In line 4, we start to train the agent by running a number of $\kappa$ episodes with our environment. Each edge server can learn to determine the placement strategy (migration or keeping the original position) of services gradually and independently after training for $\kappa$ episodes. We start to initialize environmental parameters for edge servers and users, and we generate an initial state $s_1$ in line 5. The training process in one time period $T$ starts from lines 6 to 15. For each time slot, we select an action $a_t = \mu(s_t|\theta^\mu) + \delta_t$ to determine the destination of migration by running the current policy network $\theta^\mu$ and exploration noise $\delta_t$. Since the movements of users are erratic and autonomous, we detect any migration conflicts and resolve them based on Algorithm 1 in line 8. For each user agent, we execute action $a_t$ and observe reward $r_t$ and new state $s_{t+1}$ from the environment. Then, we store the transition tuple $(s_t, a_t, r_t, s_{t+1})$ into the replay buffer $B$ in line 10. In lines 12 to 14, the actor and critic

network of the user agent will be updated according to the mini-batch of **I** transitions from *B*. In line 11, we update the critic network, which takes the state $s_t$ and action $a_t$ as input, and it outputs the action value [19]. Specifically, the critic approximates the action-value function $Q(s, a|\theta^Q)$ by minimizing the following loss function:

$$L = \frac{1}{\mathbf{I}} \sum_{w=1}^{\mathbf{I}} (r_w + \gamma Q'(s'_w, a'|\theta^{Q'}) - Q(s_w, a|\theta^Q)) \tag{16}$$

In line 12, we update the actor network, which represents the policy parameterized by $\theta$. It maximizes $\bigtriangledown_{\theta^\mu} J$ using stochastic gradient ascent, which is given by:

$$\bigtriangledown_{\theta^\mu} J \approx \frac{1}{\mathbf{I}} \sum_{w=1}^{\mathbf{I}} \bigtriangledown_a Q'(s_w, a|\theta^Q)|_{a=a_w} \bigtriangledown_{\theta^\mu} \mu(s_w|\theta^\mu) \tag{17}$$

Finally, the target network is updated by $\theta^{\mu'} \leftarrow \tau\theta^\mu + (1 - \tau)\theta^{\mu'}$ and $\theta^{Q'} \leftarrow \tau\theta^Q + (1 - \tau)\theta^{Q'}$.

---

**Algorithm 2** Dynamic service placement based on DRL

---

**Input:** Sets of edge nodes **M**, services **V**, and users **U**;
**Output:** Dynamic service placement scheme **X**;
1: Randomly initialize the actor network $\mu(s|\theta^\mu)$ and critic network $Q(s, a|\theta^Q)$ with weight $\theta^\mu$ and $\theta^Q$;
2: Initialize the target networks with weights $\theta^{\mu'} \leftarrow \theta^\mu$ and $\theta^{Q'} \leftarrow \theta^Q$;
3: Initialize replay buffer *B*;
4: **for** episode from 1 to $\kappa$ **do**
5:     Initialize environmental parameters for edge servers and users, and generate an initial state $s_1$;
6:     **for** each time slot *t* from 1 to *T* **do**
7:         Select an action $a_t = \mu(s_t|\theta^\mu) + \delta_t$ to determine the destination of migration by running the current policy network $\theta^\mu$ and exploration noise $\delta_t$;
8:         Detect migration conflicts and resolve via Algorithm 1;
9:         Execute action $a_t$ of each user agent independently, and observe reward $r_t$ and new state $s_{t+1}$ from the environment;
10:       Store the transition tuple $(s_t, a_t, r_t, s_{t+1})$ into replay buffer *B*;
11:       Randomly sample a mini-batch of **I** transitions $\{(s_w, a_w, r_w, s'_w)\}$ from replay buffer *B*;
12:       Update the critic network $Q(s, a|\theta^Q)$ by minimizing the loss function *L* in Equation (16);
13:       Update the actor network $\mu(s, a|\theta^\mu)$ by using the sampled policy gradient $\bigtriangledown_{\theta^\mu} J$ in Equation (17);
14:       Update the target networks: $\theta^{\mu'} \leftarrow \tau\theta^\mu + (1 - \tau)\theta^{\mu'}$, $\theta^{Q'} \leftarrow \tau\theta^Q + (1 - \tau)\theta^{Q'}$;
15:     **end for**
16: **end for**

---

## 5. Evaluations

In this section, we conduct extensive simulations and experiments to study the dynamic service placement problem in multiple mobile users. We develop a prototype of our framework using python, which consists of the construction of the edge network and the requests of multiple mobile users. After presenting the datasets and settings, the results are shown from different perspectives to provide insightful conclusions.

### 5.1. Basic Setting

Our prototype is built on a workstation Precision T7910 with Intel Xeon(R) E5-2620 CPU, NVIDIA RTX5000 GPU, 128 Gb memory, and a 2Tb hard disk, which runs a Linux operating system using python. We simulate our edge computing architecture based on

the campus of Beijing University of Technology with a range of $500 \times 500$ m$^2$ and set up 10 mobile edge servers in synthetic datasets, as shown in Figure 4. For each server, the setting of computing capacity randomly ranges from 20 to 25 GHz. The storage of each server ranges from 8 to 16 GB, and the bandwidth between each pair of edge servers is 0.2 GHz. We set the transmission power to be $tr = 0.5$ W and the noise power to be $N = 2 \times 10^{-3}$ [24]. In order to analyze the total delay with different numbers of users, we construct the synthetic dataset into three groups of size 20, 30, and 40. The data size of uninterrupted requests sent by users in a continuous timescale randomizes in [0.1 GB, 0.5 GB]. The settings of hyperparameters are listed in Table 2. In addition to the proposed placement algorithm, two state-of-the-art algorithms are used: dynamic service placement with no migration (DSP-NM) and dynamic service placement with all migration (DSP-AM).

- DSP-NM: Services are placed on the initialized edge server, and there is no migration in the timescale of multiple mobile users.
- DSP-AM: Services always migrate according to the users' dynamic trajectories in the timescale.

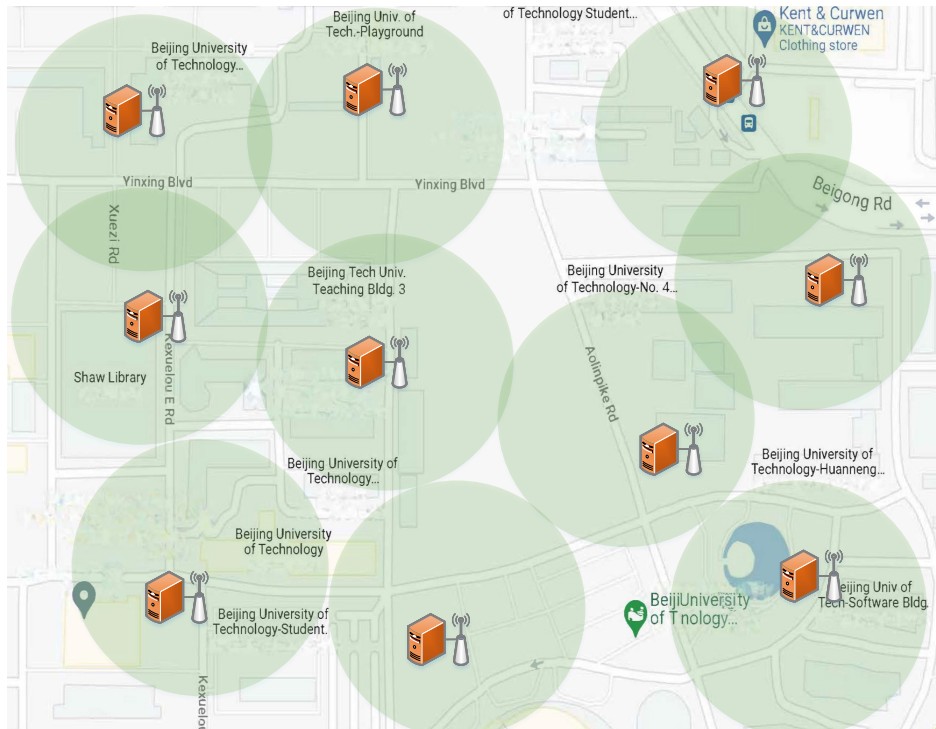

**Figure 4.** The location of servers on campus and the distribution of some users.

**Table 2.** Hyperparameter settings.

| Hyperparameter | Settings |
|---|---|
| learning rate for actor | 0.001 |
| earning rate for critic | 0.002 |
| reward decay $\gamma$ | 0.9 |
| soft replacement $\tau$ | 0.01 |
| replay memory | 200 |

### 5.2. Experiment Results

We conduct the experiments of three algorithms under different groups which are divided according to the numbers of users and the trajectories. For each group of users, we collect the results under the same settings.

5.2.1. Convergence

We investigate the convergence for three groups of mobile users (of size 20, 30, and 40), and where each user has 20 trajectories in a timescale. The results are shown in Figures 5–7. We use a black dotted line to describe the convergence trend of delay with the increasing number of iterations for each group of users. Additionally, we have the following observations. (i) For the same group of users with the same trajectory, the delay of users guided by the DSP-DRL framework is far greater than the other two comparison algorithms. As shown in Figure 5, the red and yellow lines are the results of DSP-NM and DSP-AM, which are much higher than the beginning of DSP-DRL. The relationship between the results of DSP-NM and DSP-AM is influenced by the communication delay and the migration delay, which relates to the size of users' data and the configuration files of services. In our experiment, the users are set to send data packets uninterrupted at equal time intervals. Therefore, the communication delay increases sharply when the users move frequently, resulting in a very large delay under DSP-NM. (ii) The increasing number of users has an influence on the convergence. As shown in Figures 5–7, the speed of the convergence slows down as the number of users increases. As shown in Figure 5, the total delay is close to convergence after 250 iterations. However, as shown in Figures 6 and 7, the groups with 30 and 40 users approach convergence after 400 and 420 iterations. The reason is that an increase in the number of users means a corresponding scaling in the number of services, and the probability of the migration conflict will increase, which reduces the convergence speed. (iii) The total delay fluctuates within a relatively fixed range for each group of users. Since the provisioning of edge servers is relatively dense, there exist many cross-coverage areas that provide multiple choices for users. There are many different placement results in the learning process of DSP-DRL, and the total delay generated by these results will fluctuate among several relatively fixed values during the convergence process. Therefore, the fluctuations are different under these three groups of users, which is related to the user's activity trajectories and the placement deviation of services.

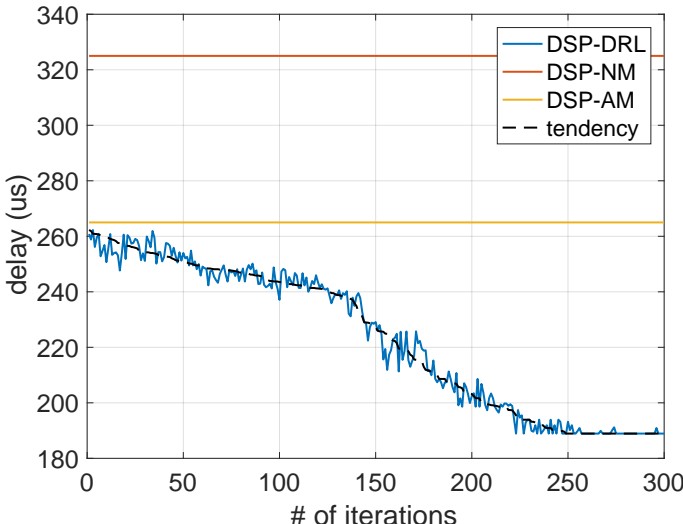

**Figure 5.** The convergence on total delay of 20 users.

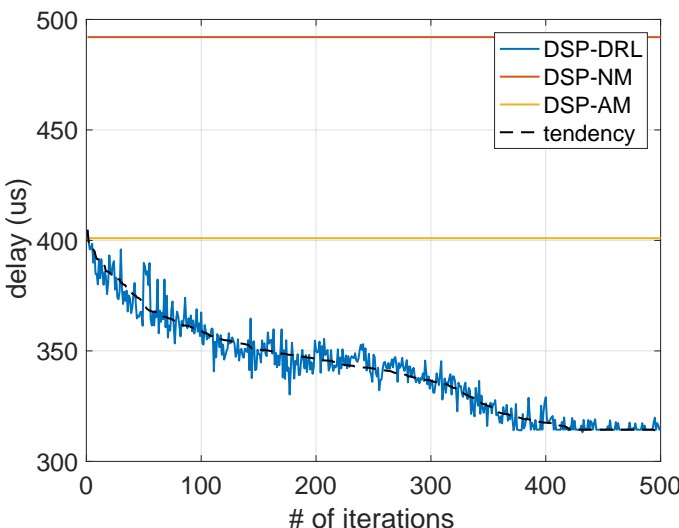

**Figure 6.** The convergence on total delay of 30 users.

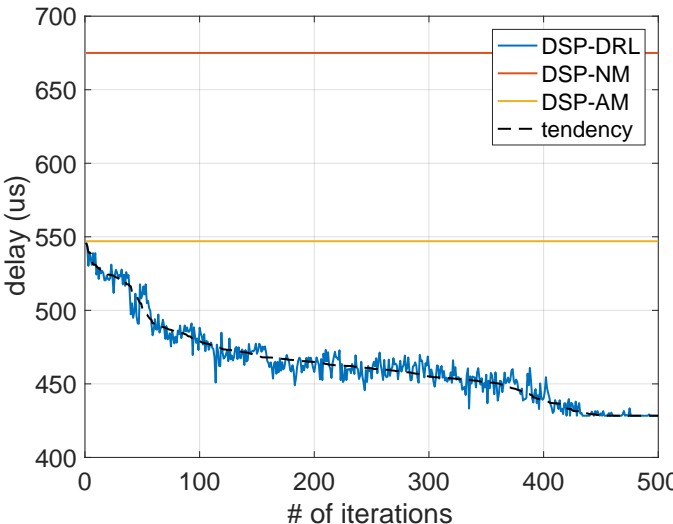

**Figure 7.** The convergence on total delay of 40 users.

5.2.2. Total Delay

According to the convergence obtained with different groups of users, we assess the average of the delay among the three groups, which are shown in Figures 8 and 9. Additionally, we have the following observations. (i) The number of users' activity trajectories collected in a timescale affects the total delay. As shown in Figure 9, the highest total delay of 40 users under DSP-NM in the case of 10 trajectories is much lower than the case of 20 trajectories. (ii) The erratic activities of end-users make the delay under these three algorithms quite different. For the users with 10 trajectories, the total delay of 20 users with DSP-NM is lower than DSP-AM. However, for groups with 30 and 40 users, the total delay of DSP-NM is higher than that of DSP-AM. For the users with 20 trajectories, the total delays under the DSP-NM of these three groups of users are all higher than DSP-AM. For both cases, DSP-DRL is always able to obtain a lower latency for different numbers of users. Compared with DSP-NM and DSP-AM, DSP-DRL can reduce the total delay by 41.2% and 32.9% under the constraints in the 10 trajectories case and 35.4% and 20.5% in the 20 trajectories case. In summary, DSP-DRL has better performance across different scales of users in mobile edge computing.

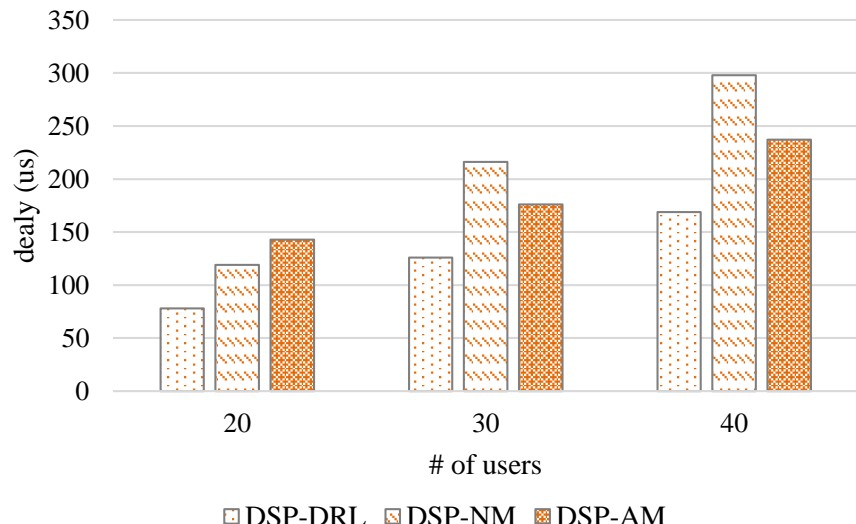

**Figure 8.** The average delay with 10 trajectories of each user group.

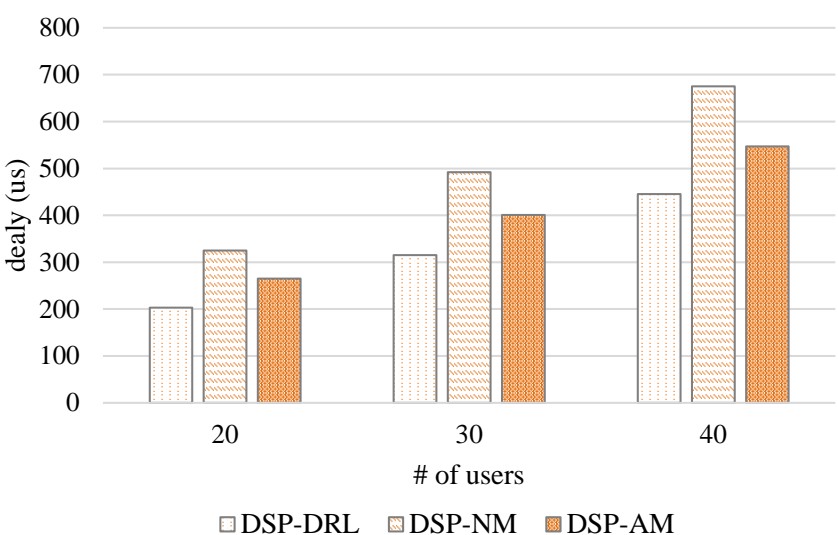

**Figure 9.** The average delay with 20 trajectories of each user group.

## 6. Conclusions

In this paper, we study the service placement problem under the continuous provisioning scenario in mobile edge computing. We propose a novel dynamic placement framework DSP-DRL based on deep reinforcement learning to optimize the total delay without overwhelming the constraints on physical resources and operational costs. In the learning framework, we propose a new migration conflict resolution mechanism to avoid the invalid state in the decision module. We formulate the service placement under the migration conflict into a mixed-integer linear programming (MILP) problem. Based on that, we propose a new migration conflict resolution mechanism to avoid the invalid state and approximate the policy in the decision module according to the introduced migration feasibility factor. Finally, we conduct extensive evaluations under various scenarios to demonstrate that our scheme outperforms existing state-of-the-art methods in terms of delay of users under the constraints on resources and cost in edge computing. For future work, we will investigate the dynamic service placement with multiple replications in mobile edge computing, in which the constraints on physical resources and consistency are also taken into consideration.

**Author Contributions:** Conceptualization, S.L. and J.W.; methodology, S.L.; software, J.S. and P.L.; validation, S.L.; formal analysis, J.W.; investigation, S.L.; resources, J.S. and P.L.; data curation, S.L. and H.L.; writing—original draft preparation, S.L.; writing—review and editing, S.L. and J.W.; supervision, J.W. and J.F.; project administration, S.L. and J.F.; funding acquisition, S.L. and H.L. All authors have read and agreed to the published version of the manuscript.

**Funding:** This research was funded by China Postdoctoral Science Foundation (2021M700366), and the Fundamental Research Foundation (040000546320508).

**Conflicts of Interest:** The authors declare no conflict of interest.

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
