# Peer review of "A Dynamic Service Placement Based on Deep Reinforcement Learning in Mobile Edge Computing"

_2673-8732, doi:10.3390/network2010008_

Round 1

Reviewer 1 Report

The proposal is reasonable, and the merit obtained by the proposed scheme is clearly shown by the experiments. These are the good aspects of the paper.

There, however, is the problem in readability. The following points are strongly recommended.

  1. Brush up the English. If possible, it is recommended to ask the English native speaker or other researcher other than the authors to check.
  2. There are some errors. For example, the definition of is “ xih(t) = 1 denote user ui using the service vh at time slot t, otherwise xih(t) = 0.”, and it is used in euation 1. By reading the equation, my understanding is that the meaning of xih(t) in the equation is whether service vh is used on the server mj or not, which slightly different for the definition. Please check if my understanding is correct or not, and if correct, make the appropriate revision. Likewise, please carry out careful check all through the paper.
  3. Revise the positioning structure of Figures, Tables, and Algorithm. It is seen that the related main body descriptions are located fur apart from these. Some of these are not placed within the appropriate sections. Algorithm 2, is partly referred in the section 3.2.2, but the full description appears later in the section 3.3.
  4. Also check That the referencing to Figure is correctly done, and check whether the things that are written in the main body sentences are correctly appear in the figures.
  5. Check whether the definition/explanation of symbol can be found near the place when it appears for the first time. There are some the definition/explanation appears fur after.
  6. Most of the references are Chinese ones. It is better to consider the balancing.

Author Response

Response summary

We would like to thank the reviewer for carefully reviewing our paper. Many constructive comments are offered. We have revised our paper according to your suggestions and provided point-to-point responses to each of your comments.

  1. Brush up the English. If possible, it is recommended to ask the English native speaker or other researcher other than the authors to check.

Response:

Thank you for offering this recommendation. We highly appreciate your constructive suggestion. We have checked our paper carefully and had it proofread by a professional proofreader. 

  1. There are some errors. For example, the definition of is “ xih(t) = 1 denote user ui using the service vh at time slot t, otherwise xih(t) = 0.”, and it is used in euation 1. By reading the equation, my understanding is that the meaning of xih(t) in the equation is whether service vh is used on the server mj or not, which slightly different for the definition. Please check if my understanding is correct or not, and if correct, make the appropriate revision. Likewise, please carry out careful check all through the paper.

Response:

Thank you very much for putting forward this question. We have carefully checked this part and found that the statement of this part needed to be improved. The definition of “xih(t)” is used to denote whether service vh is used on the server mj by user ui or not. We revised the notation “xih(t)” into “xihj(t)” and check it throughout the paper. We have already revised the definition in Session 2.1.

Let xihj(t)=1 denote user ui using the service vh which is placed on edge server mj at time slot t, otherwise xihj(t)=0.

  1. Revise the positioning structure of Figures, Tables, and Algorithm. It is seen that the related main body descriptions are located fur apart from these. Some of these are not placed within the appropriate sections. Algorithm 2, is partly referred in the section 3.2.2, but the full description appears later in the section 3.3.

Response:

We appreciate your constructive suggestion. We have repositioned the figures and algorithms.

  1. Also check That the referencing to Figure is correctly done, and check whether the things that are written in the main body sentences are correctly appear in the figures.

Response:

Thank you very much for your advice. We have checked and revised all references to figures and the corresponding main body sentences.

  1. Check whether the definition/explanation of symbol can be found near the place when it appears for the first time. There are some the definition/explanation appears fur after.

Response:

Thank you for offering this recommendation. We have checked and revised the positions of definitions/explanations.

  1. Most of the references are Chinese ones. It is better to consider the balancing.

Response:

Thank you for offering this recommendation. We have checked the references section and added some new references to keep the balance.

Reviewer 2 Report

Authors must revise your manuscript in consideration of the followings:

  • The number of section should be started with 1.
  • There is no reference of “Table 1” in the body of text.

  • There is no reference of “Figure 1” in the body of text.
  • Line 147,  you describe the notation about the set named T. In the set T, there is a member  T in T. That is making a confusion  to understand the notion.  

     t T = {0, 1, 2, ...T}

  • You make correct the sentence with a grammar error:

        "each user can only served by one service on the MEC" 

         It may be replaced "can only served" to "can only be served".

  • In section 4.2.2,  you describe  "for groups with 40 and 50 users" in line 328. But there is  no  group about "50 users" in Figure 8 and Figure 9.

Author Response

Response summary

We would like to thank the reviewer for carefully reviewing our paper. We have revised our paper according to your suggestions and provided point-to-point responses to each of your comments.

  1. The number of section should be started with 1.

Response:

Thank you for offering this recommendation, and we have revised it in our manuscript.

  1. There is no reference of “Table 1” in the body of text.

Response:

Thank you very much for putting forward this question. We have carefully checked this part and added the reference to “Table 1”.

  1. There is no reference of “Figure 1” in the body of text.

Response:

Thank you very much for your advice, and we have checked and revised all referencing to this figure and the corresponding main body sentences.

  1. Line 147, you describe the notation about the set named T. In the set T, there is a member T in T. That is making a confusion to understand the notion.

 t T = {0, 1, 2, ...T}.

Response:

We highly appreciate your constructive suggestion. We have revised this part by changing the notation of set “T”, where t ∈ T = {0, 1, 2, ...T}.

  1. You make correct the sentence with a grammar error:

        "each user can only served by one service on the MEC"

         It may be replaced "can only served" to "can only be served".

Response:

Thank you very much for your advice, and we have checked and revised it in our manuscript.

  1. In section 4.2.2, you describe "for groups with 40 and 50 users" in line 328. But there is  no  group about "50 users" in Figure 8 and Figure 9.

Response:

Thank you very much for putting forward this improper expression (groups with 30 and 40 users), we have revised it in our manuscript.

However, for groups with 30 and 40 users, the total delay of DSP-NM is higher than that of DSP-AM.

Reviewer 3 Report

This work is about a novel dynamic placement method, a new migration 8 conflicting resolution mechanism based on deep reinforcement learning to optimize the total delay on MEC case, without overwhelming the constraints on physical resources and operational costs. to avoid the invalid state. The proposed dynamic service placement framework indeed outperforms baselines in terms of efficiency and overall latency, and it is therefore has a crystal clear added value.

However, there are quite a few major points that need to be clarified more, so at this point, current work strongly needs to be reconsidered again before resubmission.

The critical points for consideration are:

  • The introduction section has no references at all. It is strongly suggested for the author to insert relative references within this section.
  • In line 70 there is no need to abruptly insert 13 references to justify just a generic statement. It is strongly suggested for the author to lessen down to a few references.
  • The author needs to specify more analytically service and coverage ranges of fig 1 in lines 151-152, as currently thius is not expressed clearly (e.g all circles seem to have the same colour).
  • The author needs to clarify the term “size of service” in line 150.
  • How, with which criteria the higher bound of max cost is deduced right after equation 5? Or, it is not necessary to explain how it has been derived for this purpose? I mean you just use it to set the Γ constraint as required for your model. Is that so?
  • The author needs to add a relative reference for deep deterministic policy gradient (DDPG) algorithm in line 176.
  • In line 187 it is clearly understandable the author implication, but it is strongly recommended to define the role of term “agent” as it is inserted in the manuscript for the first time.
  • In relation with the previous comment, agents role is defined in line 210, but It would be more proper to be placed at the point where the term agent is first referred (line 187).
  • Right after relation 12, as the author claims, why the first part of it, is fixed? The author should explain more analytically this point.
  • In line 234 the author needs to specify more the term “pre migration”. Is he referring to the training data of his model?
  • At the end of pg 10 (however there seems to be no line numbering), the author needs to specify more the sentence “For each time slot.. exploration noise dt.” especially the purpose for using parameter "exploration noise".
  • In the beginning of pg 11 the author needs to specify more the terms “actor” and “critic network” and their purpose of usage.

Author Response

Response summary

We would like to thank the reviewer for carefully reviewing our paper. We have revised our paper according to your suggestions and provided point-to-point responses to each of your comments.

  1. The introduction section has no references at all. It is strongly suggested for the author to insert relative references within this section.

Response:

Thank you very much for raising this issue. We have added the references in the introduction section.

  1. In line 70 there is no need to abruptly insert 13 references to justify just a generic statement. It is strongly suggested for the author to lessen down to a few references.

Response:

Thank you very much for raising this issue, and we have removed the references in line 70.

  1. The author needs to specify more analytically service and coverage ranges of fig 1 in lines 151-152, as currently thius is not expressed clearly (e.g all circles seem to have the same colour).

Response:

Thank you very much for putting forward this question. We have carefully checked this part and found that the description of this part needed to be improved. We have revised Figure 1 and lines 151 to 154 in Section 3.1.

Each MEC has a service range shown in Figure 1. Here, we suppose that the capacities of MECs are heterogeneous, and their service ranges are different. We use light orange color circles with different sizes to represent the coverage ranges of each MEC.

  1. The author needs to clarify the term “size of service” in line 150.

Response:

Thank you very much for putting forward this question. The term “size of service” is originally to indicate the differences in the updating delays during the service migration. We have carefully checked this term and found that the definition of “size of service” is redundant with the updating delay Υ(vi). Thus, we delete the notation |vh| in line 150 and reserve Υ(vi) in Section 3.2.3.

Υ(vi) is the updating delay of service vi, which includes service profiles transmission, rebooting software resources, and so on.

  1. How, with which criteria the higher bound of max cost is deduced right after equation 5? Or, it is not necessary to explain how it has been derived for this purpose? I mean you just use it to set the Γ constraint as required for your model. Is that so?

Response:

Thank you very much for raising this issue. In our problem formulation, we use a threshold Γ to denote the constraint of the total migration cost. The main idea of setting a constraint Γ is to avoid the expensive overhead caused by excessive migrations, however, it can be set by different service providers under the real application scenario.

6.The author needs to add a relative reference for deep deterministic policy gradient (DDPG) algorithm in line 176.

Response:

Thank you very much for putting forward this question. According to your suggestion, we have added the reference [19] for the deep deterministic policy gradient (DDPG) algorithm.

[19] Lillicrap, T. P., Hunt, J. J., Pritzel, A., Heess, N., Erez, T., Tassa, Y., & Wierstra, D. (2015). Continuous control with deep reinforcement learning. arXiv preprint arXiv:1509.02971.

  1. In line 187 it is clearly understandable the author implication, but it is strongly recommended to define the role of term “agent” as it is inserted in the manuscript for the first time. In relation with the previous comment, agents role is defined in line 210, but It would be more proper to be placed at the point where the term agent is first referred (line 187).

Response:

Thank you very much for your advice, and we have revised the description in Section 4.1.

In this paper, the objective of the agent is to realize dynamic service placement for multiple mobile users while minimizing the total delay. We first summarize the state and action spaces, reward function, and the state transition policy that are used in our reinforcement learning framework. To better describe the environment of edge servers and mobile users for the agent concisely and correctly, the state space includes the knowledge of services placed on the edge servers and the status of users that are supplied by these services. To that end, the state is designed as follows.

  1. Right after relation 12, as the author claims, why the first part of it, is fixed? The author should explain more analytically this point.

Response:

Thank you very much for putting forward this question. In this paper, we assume that each agent pre-executes the selected migration strategy during the learning stage. However, due to the insufficient capacities of some edge servers, conflicts may occur during the migration process. For edge serve mj, the total delay of users that are served by corresponding services is Zumj (t). We suppose that Zumj (t)= Z’umj (t)+ Z’’umj (t). Z’umj (t) is the delay of the services with conflicts, and Z’’umj is the delay of placed services on mj. Z’’umj is determined by the services selected by the mj edge server, which means optimizing the total delay by determining the value of (vi) to be 0 or 1, i.e., (vi) {0,1}. In this paper, we consider optimizing the service selection process during the migration conflicts occur. For set Z’umj (t), it contains all the updating delays in the set that have not migrated successfully to edge server mj. The value of Z’umj (t) is determined by the migration strategy. Z’umj (t) is 0 if the service location does not change when a conflict occurs and no migration succeeds at time slot t. Otherwise, it is a fixed value associated with the updating delay according to the migration policy. So, the value of Z’umj (t) depends on the solution of placing the conflicted services with unsuccessful migration. Therefore, Z’umj (t) is fixed compared to Z’’umj and doesn’t contribute to the total latency of edge server mj that we want to optimize.

  1. In line 234 the author needs to specify more the term “pre migration”. Is he referring to the training data of his model?

Response:

Thank you very much for putting forward this question. The pre-migration is not the training data. In this paper, we assume that each agent pre-executes the selected migration strategy during the learning stage. These services send migration requests to the target edge servers in advance and then determine whether there is a real migration by judging whether there is a conflict and how to resolve the conflict. This process is not a real migration, but hypothetical. Therefore, we use the term “pre-migration” to describe this process.

  1. At the end of pg 10 (however there seems to be no line numbering), the author needs to specify more the sentence “For each time slot.. exploration noise dt.” especially the purpose for using parameter "exploration noise".

Response:

Thank you very much for putting forward this question. In Algorithm 2, we add the exploration noise t to deal with the out-of-scope actions which may cause program exceptions. It is a function that maps the action back to the set of edge servers when the selection is not in the scope. The implementation of this part is to add randomness to action selection for exploration.

  1. In the beginning of pg 11 the author needs to specify more the terms “actor” and “critic network” and their purpose of usage.

Response:

Thank you for offering this recommendation, and we have revised the first paragraph of Section 4. In this paper, we focus on the service placement problem under the continuous provisioning scenario in mobile edge computing for multiple mobile users. Since the trajectories of mobile users are uncertain and dynamic in a timescale, the learning methods which make decisions by calculating the Q-value of the state and action are not precise. Therefore, we propose a dynamic service placement strategy based on the deep deterministic policy gradient algorithm according to the characteristic of the decision-making process. In our framework, there are two networks (main network and target network). In the main network, the critic network is used to output real-time actions for actors to implement in reality, while the actor-network is used to update the value in the network system. In the target network, they are all outputting the value of this state, but the inputs are different. The critic network will analyze the action from the actor-network plus the observation value of the state, and the actor-network will take the actor at that time. The overview of our framework is shown in Figure 2 in the manuscript.

  1. Dynamic Service Placement Framework based on Deep Reinforcement Learning

In this section, we show the detail of our novel decentralized dynamic service placement framework based on the deep reinforcement learning approach to realize the lower delay under the constraints on physical resources and costs. There are two networks (main network and target network) in our framework. In the main network, the critic network is used to output real-time actions for actors to implement in reality, while the actor network is used to update the value in the network system. In the target network, they are all outputting the value of this state, but the inputs are different. The critic network will analyze the action from the actor network plus the observation value of the state, and the actor network will take the actor at that time. Figure~\ref{fig2} shows the overall architecture of DSP-DRL framework.

Round 2

Reviewer 3 Report

This version has been imporoved, and there are no further comments therefore thiw manuscript is suggested for publications as it is